# MSM: Multi-Scale Mamba in Multi-Task Dense Prediction

## Abstract

High-quality visual representations are crucial for success in multi-task dense prediction. The Mamba architecture, initially designed for natural language processing, has garnered interest for its potential in computer vision due to its efficient modeling of long-range dependencies. However, when applied to multi-task dense prediction, it reveals inherent limitations. Unlike text processing with diverse tokenization strategies, image token partitioning requires careful consideration of multiple options. In multi-task dense prediction, each task may require specific levels of granularity in scene structure. Unfortunately, the current Mamba implementation, which segments images into fixed patch scales, fails to match these requirements, leading to sub-optimal performance. This paper proposes a simple yet effective Multi-Scale Mamba (MSM) for multi-task dense prediction. Firstly, we employ a novel Multi-Scale Scanning (MS-Scan) to establish global feature relationships at various scales. This module enhances the model's capability to deliver a comprehensive visual representation by integrating information across scales. Secondly, we adaptively merge task-shared information from multiple scales across different task branches. This design not only meets the diverse granularity demands of various tasks but also facilitates more nuanced cross-task feature interactions. Extensive experiments on two challenging benchmarks, *i.e.*, NYUD-V2 and PASCAL-Context, show the superiority of our MSM vs its state-of-the-art competitors.

## 1 Introduction

Multi-task dense prediction is a critical visual task designed to simultaneously predict outputs for various pixel-level tasks, such as semantic segmentation, depth prediction, surface normal estimation, and saliency detection. In the context of deep learning, the quality of image representation is paramount (Vandenhende et al., 2021; Crawshaw, 2020). The representation is enriched not only by extracting rich features from the input images (Xu et al., 2018; Zhang et al., 2023) but also by the synergistic interactions and complementarities among features from various tasks (Ye & Xu, 2022; Sinodinos & Armanfard, 2024). These dynamic cross-task feature interactions significantly enhance the robustness and effectiveness of the task representations in accurately capturing a wide array of visual attributes.

Initially, methodologies for multi-task dense prediction predominantly employed Convolutional Neural Networks (CNNs). These networks (Xu et al., 2018; Gao et al., 2019; Sun et al., 2021) were meticulously designed with distinct branches for each task, complemented by modules that facilitated cross-task information interactions, aiming to fortify the robustness of the representations. Nonetheless, the inherently limited receptive fields of CNN architectures frequently led to suboptimal performance. In response to these challenges, transformer-style networks (Ye & Xu, 2022; Xu et al., 2023) demonstrate exceptional proficiency in modeling long-range dependencies. This capability substantially improves the representational effectiveness of models in handling the complexities of multi-task scenarios. the computational complexity of attention mechanisms, which increases quadratically with the resolution, presents a substantial challenge for multi-task dense prediction. To mitigate this limitation, researchers (Bhattacharjee et al., 2022; 2023; Jiang et al., 2024) have adopted the Swin Transformer (Liu et al., 2021) as the foundational framework for implementing window-based attention to reduce computational demands. However, when implementing task feature refinement in the decoder, this strategy greatly limits the scope of cross-task interaction,

Figure 1: Comparison of task attention with (w/ MS) and without MS-Scan (w/o MS). Our approach demonstrates superior alignment between scene structural relationships and task-specific requirements across all tasks.

contradicting the original objective of task interaction, which is to extract as much valuable information as possible. Therefore, how to enhance global modeling capability while maintaining reduced computational cost remains an unresolved issue.

Recently, with linear complexity in long-range dependency modeling, Mamba (Gu & Dao, 2024; Dao & Gu, 2024) has excelled in natural language processing and demonstrated the potential in visual tasks (Liu et al., 2024; Ma et al., 2024). Inspired by these, MTmamba (Lin et al., 2024) replaced window-based attention with the Mamba module in decoder stage, thereby enhancing representation quality, which combines Self-Task Mamba (STM) block and Cross-Task Mamba (CTM) block to facilitate cross-task information exchange and model long-range dependencies. However, they overlooked the gap between the fixed tokenization in Mamba processing and the requirement for representation diversity in multi-task dense prediction. Specifically, Mamba processes features by converting them into sequences of tokens, which is more complex for images than text. Unlike text, where multiple tokenization strategies are viable, image tokenization (patches) requires careful consideration of diverse options. And this is crucial in multi-task dense prediction, due to each task may have varying requirements of granularity in scene representation. Unfortunately, the current implementation of MTMamba, which segments images into fixed patches, will propagate shared information at the same granularity, consequently resulting in sub-optimal performance.

To address these challenges, we propose a simple yet effective Multi-Scale Mamba (MSM) method. MSM is an extension of the existing MTMamba (Lin et al., 2024) approach with a task-aware hierarchical scene modeling function, which improves the adaptability of individual task representations, as shown in Figure 1. Specifically, we introduce a novel Multi-Scale Scanning (MS-Scan) to deliver a comprehensive visual representation. Based on the MS-Scan mechanism, we developed the Task-Specific Multi-Scale Mamba (TS-MSM) module and the Cross-Task Multi-Scale Mamba (CT-MSM) module. In the TS-MSM module, features are initially partitioned into multiple spaces, where the scene structure of images is modeled at various scales. Subsequently, specific tasks integrate multi-scale scene structural information to enhance task-specific representations as required. Within the CT-MSM module, we first consolidate all task representations and extract hierarchical task-shared scene structural information. Following that, different task branches adaptively merge task-shared representations from multiple scales to accommodate the varying demands of different tasks for image structural granularity.

The main contributions of this study are summarized as follows:

- We propose MSM for multi-task dense prediction, featuring a novel MS-Scan at its core to alleviate the difficulty of feature learning in multi-task dense scene prediction.

- We design a TS-MSM module and a CT-MSM module. these modules enhance the model's capability to deliver a comprehensive visual representation and meet the diverse granularity demands of tasks.

- Extensive experiments on two multi-task dense prediction benchmarks (i.e. PASCAL Context and NYUD-v2) verify the effectiveness of the proposed method, which demonstrates superior performance compared with the previous state-of-the-art methods.

## 2 RELATED WORK

**Multi-Task Learning.** Most existing multi-task learning works have primarily focused on optimizing training processes and designing network structures. The optimization approaches can be categorized into gradient manipulation (Yu et al., 2020; Navon et al., 2022; Jeong & Yoon, 2024; Ye et al., 2024) and loss-balancing (Chen et al., 2018; Kendall et al., 2018) to coordinate resource allocation for various tasks during training. The approaches of structural design endeavor to enhance task representation learning by devising various mechanisms. Some CNN-based methods manually design interaction mechanisms to extract useful information across tasks. For example, using intermediate auxiliary tasks (Xu et al., 2018) or designing distillation methods (Vandenhende et al., 2020) to fuse encoder features from multiple stages. With the advancement of Transformer, current methods have gained improved global task interaction capabilities to enhance task representation efficiency. Certain approaches (Bhattacharjee et al., 2022; Shoouri et al., 2023) employ pairwise interactions through the selection of a reference task, whereas others (Ye & Xu, 2022; Xu et al., 2023; Ye & Xu, 2023; Li et al., 2024) facilitate global interactions across all tasks. To mitigate computational complexity, MTMamba (Lin et al., 2024) introduced Mamba (Gu & Dao, 2024) to multi-task learning, which showcases effective long sequence modeling capabilities and achieving satisfactory performance. However, it overlooked the different requirements of scene structure granularity for various tasks in cross-task interaction.

**State Space Models.** In efficient long-range dependency modeling methods, state space models (Gu et al., 2021b; Smith et al., 2022) has become a striking alternative to Transformers. (Gu et al., 2021a) proposed a Structured State Space Sequence (S4) model based on a new parameterization, which alleviates the computational and memory efficiency issues faced by SSM. Subsequently, numerous efforts (Fu et al., 2023; Mehta et al., 2022) are dedicated to bridging the performance disparity between SSMs and Transformers. For example, H3 (Fu et al., 2023) proposed a new SSM layer to bridge the gap between performance and computational efficiency. Mamba (Gu & Dao, 2024) introduced an input-based parameterization method and hardware-aware algorithm, achieving performance on par with Transformers in natural language processing. This success has spurred various endeavors (Zhu et al., 2024) towards Mamba's adaptation for visual tasks.

The preservation of comprehensive image structural information poses a critical challenge in Mamba's sequential processing model, which has attracted considerable attention and effort (Zhu et al., 2024; Liu et al., 2024; Yang et al., 2024; Huang et al., 2024; Zhao et al., 2024). Vision Mamba (Zhu et al., 2024) introduces a novel bidirectional Mamba block (Vim) that annotates image sequences by embedding positional information, employing a bidirectional state space model to compress visual representations. Additionally, VMamba (Liu et al., 2024) proposes the 2D Selective Scan (SS2D), a four-way scanning mechanism tailored for spatial domain traversal, aimed at enhancing Mamba's image modeling capabilities. Subsequent research studies have explored various scan patterns and combinations tailored to different tasks or scenarios (Yang et al., 2024; Huang et al., 2024; Zhao et al., 2024). However, by utilizing fixed token sizes, these methods overlook the importance of hierarchical spatial structural information in visual tasks.

## 3 MAIN METHOD

We first outline the overall architecture of Multi-Scale Mamba for multi-task dense prediction in Section 3.1, then delve into the Multi-Scale Mamba Decoder and Multi-Scale Scan in Section 3.2 and 3.3 respectively, followed by a discussion of the optimization objectives in Section 3.4.

### 3.1 PINELINE OF MULTI-TASK DENSE PREDICTION

Similar to previous approaches (Bhattacharjee et al., 2022; Zhang et al., 2023; Lin et al., 2024), our MSM for multi-task dense prediction consists of two main components: a task-shared encoder $\Phi$ for extracting task-generic representations and a decoder $\Theta$ for refining features and generating predictions for individual tasks, as illustrated in Figure 2(a). This can be formulated as:

$$\hat{\mathbf{Y}} = \{\hat{\mathbf{Y}}_1, \hat{\mathbf{Y}}_2, \dots, \hat{\mathbf{Y}}_T\} = \Theta \circ \Phi(I), \tag{1}$$

where $I \in \mathbb{R}^{H \times W \times 3}$ denotes the RGB input, $\hat{\mathbf{Y}}_t$ represents the prediction for task $t$ with the same height $H$ and width $W$ as $I$, and $T$ denotes the total number of tasks. The decoder is the key

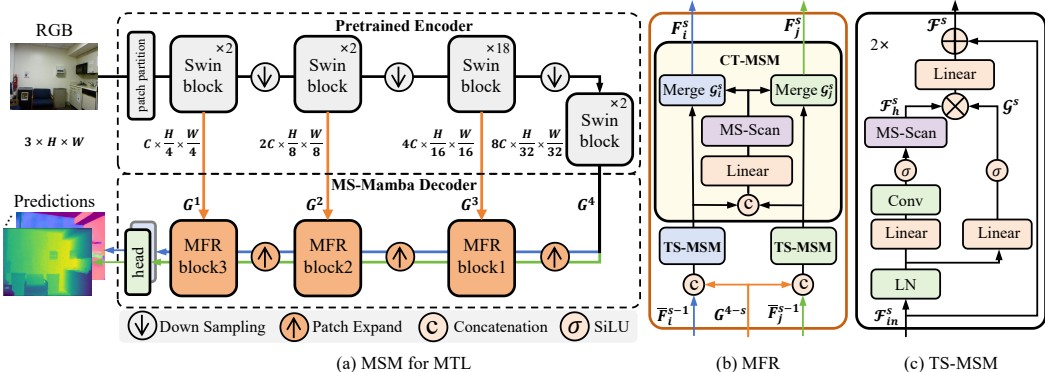

Figure 2: Framework of the proposed MSM for multi-task dense prediction. (a) overall of MSM, illustrating with depth estimation and surface normal estimation tasks. (b) Details of the MFR block, which include T task-specific TS-MSM blocks and a task-shared CT-MSM block. (c) Details of TS-MSM, the core component MS-Scan is illustrated in Figure 3.

component of our method and will be described in detail in the following section. Here, we first introduce the encoder.

The encoder shares similarities with other methods (Lin et al., 2024). We utilized a pretrained Swin Transformer (Liu et al., 2021) to extract task-generic features, which begins by dividing the input image $I \in \mathbb{R}^{H \times W \times 3}$ into $H/w \times W/w$ tokens of dimension $C$ through patch partitioning and linear layers, where $w$ denotes the partition size. These tokens are then processed through multiple stages involving alternating patch merging and Swin Transformer block processing, ultimately yielding hierarchical image representations:

$$\mathbf{G} = \{\mathbf{G}^1, \mathbf{G}^2, \mathbf{G}^3, \mathbf{G}^4\} = \Phi(I), \quad \mathbf{G}^i \in \mathbb{R}^{C_i \times H_i \times W_i}, \tag{2}$$

where $\mathbf{G}$ represents the task-generic features extracted from the encoder $\Phi$. In our practical implementation, we utilize a partition size of 4, resulting in the following shapes for $\mathbf{G}$: $C \times \frac{H}{4} \times \frac{W}{4}, 2C \times \frac{H}{8} \times \frac{W}{8}, 4C \times \frac{H}{16} \times \frac{W}{16}$ and $8C \times \frac{H}{32} \times \frac{W}{32}$, respectively.

## 3.2 MULTI-SCALE MAMBA DECODER

As the core of the proposed MSM Model, the Multi-Scale Mamba Decoder consists of three Multi-Scale Mamba Feature Refinement (MFR) blocks and $T$ task heads, as depicted in Figure 2(a). This architecture refines the task-generic features $\mathbf{G}$ obtained from the encoder into task-specific features $\mathbf{F}$, which are crucial for generating task predictions $\hat{\mathbf{Y}}$. The task-specific features are represented as $\mathbf{F} = \{\mathbf{F}_t\}_{t=1}^T$, where each $\mathbf{F}_t$ is defined as follows:

$$\mathbf{F}_t = \{\mathbf{F}_t^1, \mathbf{F}_t^2, \mathbf{F}_t^3\}, t \in \{1, 2, \ldots, T\}, \tag{3}$$

where $\mathbf{F}_t$ comprises three representations that correspond to the first three encoder stages, with dimensions of $4C \times \frac{H}{16} \times \frac{W}{16}, 2C \times \frac{H}{8} \times \frac{W}{8}$, and $C \times \frac{H}{4} \times \frac{W}{4}$, respectively. Finally, the last refined features $\{\mathbf{F}_t^3\}_{t=1}^T$ are input into the task heads to produce the final predictions $\{\hat{\mathbf{Y}}_t\}_{t=1}^T$.

The proposed MFR block in the decoder is designed to bridge the gap between task-generic and task-specific representations, as illustrated in Figure 2(b). To meet the varying demands for scene structure granularity across different tasks, especially during task interactions, we introduce two specialized multi-scale Mamba modules: TS-MSM and CT-MSM. For the $s$-th MFR block, the input is derived from two sources: (1) task-generic features $\mathbf{G}^{4-s}$ from the corresponding encoder stage, and (2) $\bar{\mathbf{F}}^{s-1}$, which is obtained by expanding the fine-tuned features $\mathbf{F}^{s-1} = \{\mathbf{F}_t^{s-1}\}_{t=1}^T$ from the preceding MFR block. For the first MFR block, $\mathbf{G}^4$ is replicated $T$ times, substituting for $\mathbf{F}^{s-1}$ as input for each task. During MFR processing, task-generic features $\mathbf{G}^{4-s}$ are initially concatenated with the expanded task-specific features $\bar{\mathbf{F}}_t^{s-1}$ within each task branch. This combined

input then undergoes processing through the TS-MSM and CT-MSM modules to yield the refined features $\mathbf{F}^s$. For clarity, we will utilize a superscript $s$ to denote processing in the $s$-th MFR block in the subsequent sections.

**Taks-Specific Mluti-Scale Mamba Block.** The TS-MSM primarily aims to construct comprehensive representations through task-internal interactions. Its architecture is illustrated in Figure 2(c) and comprises two main branches: the scan branch and the gating branch. In the scan branch, we integrate local information using convolutional layers and activation functions, followed by the implementation of a novel Multi-Scale Scan mechanism (MS-Scan) to derive a hierarchical global scene structure representation $\mathcal{F}_h^s$. Simultaneously, in the gating branch, we generate a gating signal $\mathcal{G}^s$ using an activation function to regulate the flow of information within the scan branch. Subsequently, we adjust the channel dimensions of the multi-scale scene representation by applying a linear projection $\mathcal{P}$ and establish a residual connection between $\mathcal{F}_h^s$ and the input $\mathcal{F}_{in}^s$. This process is repeated twice to produce the final output $\mathcal{F}^s$:

$$\mathcal{F}^s = \mathcal{F}_{in}^s + \mathcal{P}(\mathcal{F}_h^s \times \mathcal{G}^s). \tag{4}$$

**Cross-Task Mluti-Scale Mamba Block.** The CT-MSM is designed to address the varying demands for scene granularity across tasks during task interactions. As illustrated in the upper portion of Figure 2(b), it begins by concatenating features $\{\mathcal{F}_t^s\}_{t=1}^T$ from different task branches to construct multi-scale task-shared features $\mathcal{F}_{ms}^s$ using the Multi-Scale Scan (MS-Scan) mechanism. Subsequently, each task branch adaptively merges $\mathcal{F}_{ms}^s$ using the Merge operation to obtain finely-tuned task representations $\mathbf{F}^s = \{\mathbf{F}_t^s\}_{t=1}^T$:

$$\mathbf{F}_t^s = \mathrm{Merge}_t \circ \Psi \circ \mathcal{P}([\mathcal{F}_1^s, \mathcal{F}_2^s, \ldots, \mathcal{F}_T^s]), t \in \{1, 2, \ldots, T\}, \tag{5}$$

where $[\cdot, \cdot]$ denotes channel-wise concatenation, $\mathcal{P}$ represents a linear projection, $\Psi$ is the MS-Scan block (which will be elaborated in the following Section 3.3), and $\mathrm{Merge}_t$ is the feature fusion method in MTMamba (Lin et al., 2024) for task $t$, where $\mathcal{F}_t^s$ is first processed through convolution, SS2D (Liu et al., 2024) and sigmoid function to generate the selection value $\mathcal{G}_t^s$. The final fused feature is obtained by weighting $\mathcal{F}_{ms}^s$ and $\mathcal{G}_t^s$ on $\mathcal{G}_t^s$, the detail is described in Appendix A.1.

**Task Head.** After obtaining the final refined task representations $\mathbf{F}^S$ from the last MFR block, each task employs a task-specific head to produce the final output. We incorporate an expansion layer alongside a linear projection:

$$\hat{\mathbf{Y}}_t = \mathcal{P} \circ \mathrm{Expand}(F_t^S), t \in \{1, 2, \ldots, T\}, \tag{6}$$

where Expand denotes a module designed to double the feature resolution $H$ and $W$, consisting of a linear projection followed by a reshape operation. The operator $\mathcal{P}$ represents a linear layer that projects the feature channels to the required number of channels specific to each task.

## 3.3 Multi-Scale Scan

Mamba processes features by converting them into sequences of tokens. While a variety of tokenization strategies exist for text, image tokenization requires careful consideration of multiple approaches. Previous research has demonstrated that multi-scale processing is particularly effective for image data. To leverage this advantage, we propose a multi-scale scan mechanism that serves as a cornerstone of MSM model, as illustrated in Figure 3. In this framework, we employ multiple scanning scales, denoted as $\{s_i\}_{i=1}^N$, in multiple branches $\{\mathcal{B}_i\}_{i=1}^N$, and transform the input image feature $x \in \mathbb{R}^{C \times H \times W}$ into token sequences of varying dimensions for Mamba modeling. For instance, the initial image feature can be tokenized into a sequence with a total length of $H \times W$, where each token has a dimension of $C$, represented as $C \times (H \times W)$. When applying a scanning scale of $s_i = 2$, the image is divided into non-overlapping patches, resulting in a tokenized feature sequencewith a dimensionality of $4C \times \left(\frac{H}{2} \times \frac{W}{2}\right)$. Specifically, MS-Scan comprises three key components: input handling, multi-scale scanning, and multi-scale fusion.

**Input Handling.** To construct inputs for $N$ different scanning branches $\mathcal{B} = \{\mathcal{B}_i\}_{i=1}^N$, we perform two main operations. **(1)** Channel Split ($\mathcal{S}$): We begin by splitting the input representation $x$ into $N$ sub-features $\{x_i\}_{i=1}^N \in \mathbb{R}^{m \times H \times W}$ along the channel dimension, where $m = C/N$. **(2)** Window Tokenization ($\mathcal{W}_i$): For the $i$-th branch $\mathcal{B}_i$ with scan scale $s_i$, we first divide $x_i$ into $\frac{H}{s_i} \times \frac{W}{s_i}$ non-overlapping patches, each of size $m \times s_i \times s_i$. Subsequently, we concatenate the pixel feature

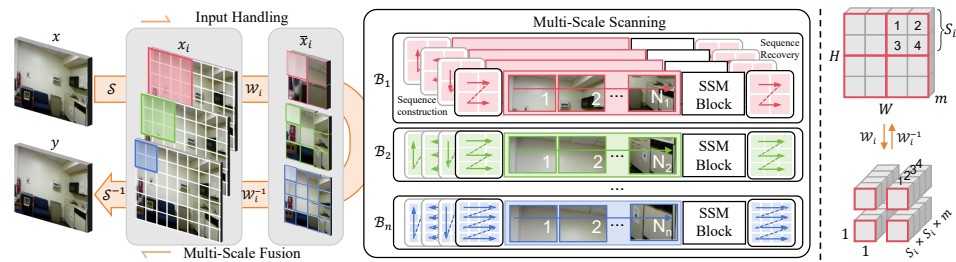

Figure 3: Left: Instructions for MS-Scan. It consists of three distinct operations, Input Handling, Multi-Scale Scanning, and Multi-Scale Fusion. Right: Illustration of Window Tokenization ($\mathcal{W}_i$) and Token Windowing ($\mathcal{W}_i^{-1}$) with a scan scale of 2.

values in each patch along the channel dimension, resulting in the scan input $\bar{x}_i$ with a shape of $(m \times s_i \times s_i) \times \frac{H}{s_i} \times \frac{W}{s_i}$. Ultimately, we obtain the input for all branches:

$$\{\bar{x}_1, \bar{x}_2, \ldots, \bar{x}_N\} = \{\mathcal{W}_1, \mathcal{W}_2, \ldots, \mathcal{W}_N\} \circ \mathcal{S}(x). \tag{7}$$

**Multi-Scale Scanning.** Following the input handling process, we employ distinct scanning scales to construct a multi-scale scene representation in each branch. For all branches $\mathcal{B}$, we utilize the four-way scanning method (SS2D) from VMamba (Liu et al., 2024) to generate scene features at the specified scale. This method creates four token sequences, each shaped as $C_i \times (H_i \times W_i)$, by scanning the input features $\bar{x}_i \in \mathbb{R}^{C_i \times H_i \times W_i}$ in four directions. The resulting sequences are then processed by SSM (Gu & Dao, 2024) and combined to produce the output feature $\bar{y} = \{\bar{y}_i\}_{i=1}^{N}$:

$$\bar{y}_i = \text{SS2D}(\bar{x}_i) \in \mathbb{R}^{C_i \times H_i \times W_i}. \tag{8}$$

**Multi-Scale Fusion.** In our approach to multi-scale feature fusion, we adopt a methodology that reverses the input handling process, consisting of two key steps. **(1)** Token Windowing ($\mathcal{W}_i^{-1}$): For each branch $\mathcal{B}_i$, we split each pixel feature into $s_i \times s_i$ segments along the channel dimension: $\{\bar{y}_{i,j}\}_{j=1}^{s_i \times s_i} \in \mathbb{R}^{m \times 1 \times 1}$ These segments are then concatenated along the spatial dimensions (height and width) to form patches, which are subsequently combined to produce the output for $\mathcal{B}_i$. **(2)** Channel Concatenation ($\mathcal{S}^{-1}$): We concatenate the features from all branches along the channel dimension, yielding the final output feature $y \in \mathbb{R}^{C \times H \times W}$:

$$y = \mathcal{S}^{-1} \circ \{\mathcal{W}_1^{-1}, \mathcal{W}_2^{-1}, \ldots, \mathcal{W}_n^{-1}\}(\bar{y}_1, \bar{y}_2, \ldots, \bar{y}_n), \tag{9}$$

where $\mathcal{S}^{-1}$ and $\mathcal{W}_i^{-1}$ refer to the inverse operation of $\mathcal{S}$ and $\mathcal{W}_i$ respectively.

### 3.4 OPTIMIZATION OBJECTIVE

We jointly train all tasks to optimize muti-scale mamba decoder $\Theta$ and task-shared encoder $\Phi$. To maintain consistency with previous approaches, we use $L1$ loss for depth estimation and surface normal estimation tasks and the cross-entropy loss for other tasks, therefore, the optimization objective can be expressed as follows:

$$L = \sum_{t_i \in \mathcal{T}} \lambda_t \mathcal{L}_t(\Theta \circ \Phi(I), \mathbf{Y}_t), \tag{10}$$

where $\mathcal{T}$ is the set of all tasks, $\lambda_t$, $\mathcal{L}_t$ and $\mathbf{Y}_t$ are the loss weight, loss function, and task label for image $I$ in task $t$ respectively.

## 4 EXPERIMENTS

### 4.1 EXPERIMENTAL SETUP

**Datasets.** We performed experiments using the benchmark datasets NYUDv2 (Silberman et al., 2012) and PASCAL Context (Chen et al., 2014). NYUDv2 primarily focuses on indoor scenes,

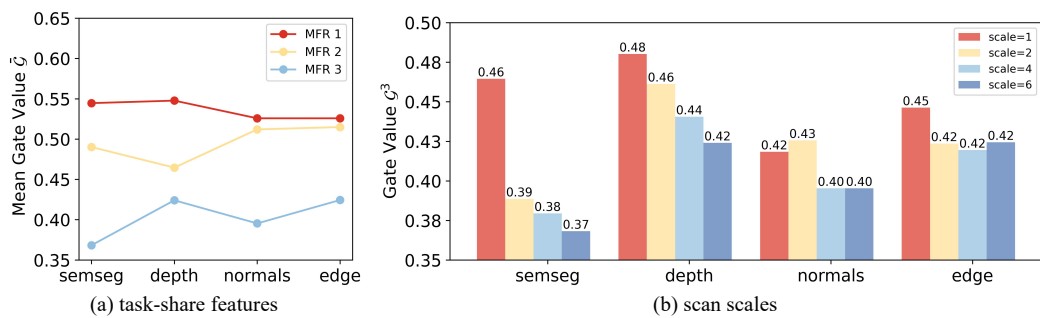

Figure 4: (a) Preference for task-share features in MFR blocks. (b) Preference for different scan scales in the final MFR block.

with 795 and 654 RGB images for training and testing purposes. Tasks in NYUDv2 include 40-class semantic segmentation, monocular depth estimation, surface normal estimation, and object boundary detection. PASCAL Context encompasses indoor and outdoor scenes, offering pixel-level labels for tasks like semantic segmentation, human parsing, object boundary detection, surface normal estimation, and saliency detection tasks. This dataset contains 4,998 training images and 5,105 test images.

**Implementation Details.** We employ a pretrained Swin-Large Transformer (Liu et al., 2021) on ImageNet-22K (Deng et al., 2009) as our encoder. Our models are trained on the NYUD-v2 dataset for 50,000 iterations with a batch size of 4, and on the Pascal Context dataset for 75,000 iterations with a batch size of 6. Across all datasets, we use the Adam optimizer with a learning rate of $5 \times 10^{-5}$ and a weight decay rate of $1 \times 10^{-5}$, alongside a polynomial learning rate scheduler. The preliminary decoder has an output channel number of 768. We follow common practice (Ye & Xu, 2022; Lin et al., 2024) in resizing the input images and applying data augmentation techniques. Specifically, we resize the input images of NYUDv2 and PASCAL-Context to $448 \times 576$ and $512 \times 512$, respectively, and apply random color jittering, random cropping, random scaling, and random horizontal flipping.

**Evaluation Metrics.** Mean Intersection over Union (mIoU) is employed for semantic and human parsing tasks. Root Mean Square Error (RMSE) and mean angle error (mErr) are used for depth and surface normal estimation tasks respectively. Saliency detection tasks utilize maximal F-measure (maxF), and object boundary detection tasks use optimal-dataset-scale F-measure (odsF). We also use the multi-task gain $\Delta_{MTL}$ (Vandenhende et al., 2021) evaluate the overall task performance.

## 4.2 EXPERIMENTAL RESULTS

**Statistical Analysis of Task Preferences.** We analyzed the task preferences for the multi-scale task-share feature across different decoding stages. Specifically, we conducted statistics in three MFRblocks in the decoder, which were set to scan scales of $\{1, 2\}$, $\{1, 2, 4\}$, and $\{1, 2, 4, 6\}$, respectively. In Figure 4 (a), we calculated the mean of the task-share feature selection value $\bar{\mathcal{G}}_t$ in CT-MSM across all scan scales in the module and averaged the results over all test images. The results indicate that as the network depth increases, the specialization of features for each task is enhanced, thereby reducing the demand for task-share representation. In Figure 4 (b), we compared the selection value $\mathcal{G}_t^3$ across four scan scales at the last MFR block. There are significant differences among tasks within the same cross-task interaction stage. These results indicate that meeting the requirements of different tasks for scene structure granularity is crucial in the interaction process.

**Comparison with State-of-the-art Methods.** Table 1 and Table 2 report a comparison of the proposed MSM against previous state-of-the-art methods, including MTmamba (Lin et al., 2024), MQ-Transformer (Xu et al., 2023), InvPT (Ye & Xu, 2022), ATRC (Brüggemann et al., 2021), MTI-Net (Vandenhende et al., 2020), PAD-Net (Xu et al., 2018), PSD (Zhang et al., 2019), PAP (Zhou et al., 2020), Cross-Stitch (Misra et al., 2016) and ASTMT (Maninis et al., 2019) on NYUD-V2 and PASCAL-Context dataset respectively. Notably, the previous best method, *i.e.*, MTmamba, and our MSM are built upon the Transformer-encoder Mamba-decoder architecture with the same back-

Table 1: Quantitative comparison of different methods on NYUD-v2 dataset.

| Model | Semseg mIoU ↑ | Depth RMSE↓ | Normal mErr ↓ | Boundary odsF ↑ |
|---|---|---|---|---|
| *CNN based* | | | | |
| Cross-Stitch | 36.34 | 0.6290 | 20.88 | 76.38 |
| PAP | 36.72 | 0.6178 | 20.82 | 76.42 |
| PSD | 36.69 | 0.6246 | 20.87 | 76.42 |
| PAD-Net | 36.61 | 0.6270 | 20.85 | 76.38 |
| MTI-Net | 45.94 | 0.5365 | 20.27 | 77.86 |
| ATRC | 46.33 | 0.5363 | 20.18 | 77.94 |
| *Transformer based* | | | | |
| InvPT | 53.66 | 0.5183 | 19.04 | 78.10 |
| MQTransformer | 54.84 | 0.5325 | 19.67 | 78.20 |
| *Mamba based* | | | | |
| MTMamba | 55.82 | 0.5066 | 18.63 | 78.70 |
| **Ours** | **57.79** | **0.4832** | **18.63** | **79.00** |

Table 2: Quantitative comparison of different methods on Pascal-Context dataset.

| Model | Semseg mIoU ↑ | Parsing mIoU ↑ | Saliency maxF↑ | Normal mErr ↓ | Boundary odsF ↑ |
|---|---|---|---|---|---|
| *CNN based* | | | | | |
| PAD-Net | 53.60 | 59.60 | 65.80 | 15.30 | 72.50 |
| ASTMT | 68.00 | 61.10 | 65.70 | 14.70 | 72.40 |
| MTI-Net | 61.70 | 60.18 | 84.78 | 14.23 | 70.80 |
| ATRC | 62.69 | 59.42 | 84.70 | 14.20 | 70.96 |
| ATRC-ASPP | 63.60 | 60.23 | 83.91 | 14.30 | 70.86 |
| ATRC-BMTAS | 67.67 | 62.93 | 82.29 | 14.24 | 72.42 |
| *Transformer based* | | | | | |
| InvPT | 79.03 | 67.61 | **84.81** | 14.15 | 73.00 |
| MQTransformer | 78.93 | 67.41 | 83.58 | 14.21 | 73.90 |
| *Mamba based* | | | | | |
| MTmamba | 81.11 | 72.62 | 84.14 | 14.14 | 78.80 |
| **Ours** | **81.38** | **72.87** | 84.41 | **14.13** | **78.83** |

bone. On NYUD-v2, the performance of Semseg is clearly boosted from the previous best, *i.e.*, 55.82 to 57.79 (**+1.97**). on Pascal-Context, we achieved superior performance on all tasks compared to MTMamba.

**Effectiveness of TS-MSM and CT-MSM.** We performed ablation experiments on TS-MSM and CT-MSM using the NYUD-V2 dataset. These experiments all used Swin-Large Transformer as encoder. The term "single-task" indicates that each task possesses its task-specific model, utilizing two particular Swin Transformer blocks in each stage of the decoder, "baseline" denotes MT-Mamba, "TS-MSM only" denotes only equipped baseline with TS-MSM, "CT-MSM only" denotes only equipped baseline with CT-MSM, and "ST-MSM+CT-MSM" is the default method of MSM. The results presented in Table 3 highlight the essential role of satisfying the diverse demands for scene structural granularity in task interaction. Furthermore, the application of MS-scan during task-internal feature refinement in TS-MSM has been shown to significantly improve performance, showcasing the advantages of the MSM design.

Table 3: Effectiveness of ST-MSM and CT-MSM on NYUDv2 dataset.

| Model | Semseg mIoU ↑ | Depth RMSE↓ | Normal mErr ↓ | Boundary odsF ↑ | MTL Gain $\Delta_m$ ↑ |
|---|---|---|---|---|---|
| STL Model | 54.32 | 0.5166 | 19.21 | 77.30 | +0.00 |
| Baseline | 55.82 | 0.5066 | 18.63 | 78.70 | +2.38 |
| TS-MSM only | 56.89 | 0.4840 | 18.68 | 78.80 | +3.93 |
| CT-MSM only | 57.13 | 0.4822 | 18.67 | 78.80 | +4.14 |
| ST-MSM+CT-MSM | **57.79** | **0.4832** | **18.63** | **79.00** | **+4.51** |

**Ablation Study on Scan Scales.** We experimented with the impact of varying scan scales on model performance, as shown in Figure 5. Experiments were conducted on NYUDv2 dataset with Swin-Large Transformer as encoder. We experimented with three different settings, **Type 1**: all MFRs use {1,2} two scan scales; **Type 2**: three MFRs utilize {1,2}, {1,4}, {1,6} respectively; **Type 3**: all MFRs use {1,4} two scan scales. The results showed that in multi-scale scanning, all scale divisions achieved better performance than single scale, *i.e.*, MTMamba (mark with dashed lines in Figure 5). Ultimately, we adopt the 1,4 scale setting for all MFRs as the final configuration for MSM.

**Ablation Study on Scan Numbers.** We conduct ablation experiments on the impact of varying scan numbers, as shown in Table 4. We compared four different settings: (**1**) all three MFRs use {1} scan scale; (**2**) three MFRs use {1,2}, {1,4}, and {1,6} respectively; (**3**) all MFRs use {1,2,4}; (**4**) three MFRs use {1,2}, {1,2,4} and {1,2,4,6} respectively. The results showed that employing appropriate scan scale partitions can effectively enhance overall performance. Significantly, all variations in scan quantity settings yielded notable performance enhancements.

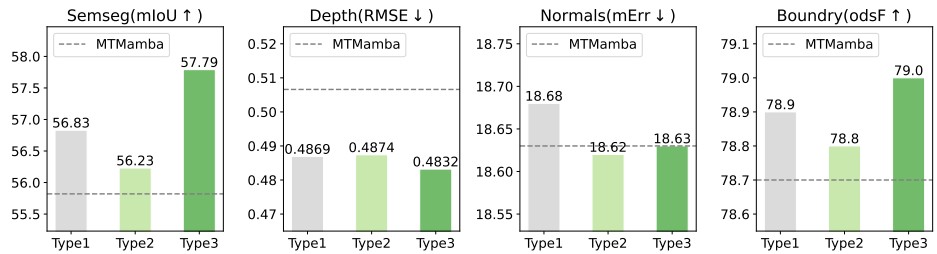

Figure 5: Performance comparison of different scan scale settings in MSM.

Table 4: Different scan numbers.

| | Scan Scale | | Semseg | Depth | Normal | Boundary | MTL Gain |
|---|---|---|---|---|---|---|---|
| MFR 1 | MFR 2 | MFR 3 | mIoU ↑ | RMSE↓ | mErr ↓ | odsF ↑ | $\Delta_m$ ↑ |
| {1} | {1} | {1} | 55.82 | 0.5066 | 18.63 | 78.70 | +2.38 |
| {1,2,4} | {1,2,4} | {1,2,4} | 56.63 | 0.4878 | 18.83 | 78.70 | +3.40 |
| {1,2} | {1,2,4} | {1,2,4,6} | 56.93 | 0.4850 | 18.52 | 78.90 | +4.15 |
| {1,4} | {1,4} | {1,4} | **57.79** | **0.4832** | **18.63** | **79.00** | **+4.51** |

Table 5: Different encoders.

| Model | Semseg mIoU ↑ | Depth RMSE↓ | Normal mErr ↓ | Boundary odsF ↑ |
|---|---|---|---|---|
| MTMamba-Base | 53.62 | 0.5126 | 19.28 | 77.70 |
| MSM-Base | 54.64 | 0.5038 | 19.03 | 78.10 |
| MTMamba-Large | 55.82 | 0.5066 | 18.63 | 78.70 |
| MSM-Large | **57.79** | **0.4832** | **18.63** | **79.00** |

**Performance on Different Encoder.** We evaluate the effect of model size on experimental performance, presented in Table 5. All experiments were conducted on the NYUDv2 dataset. We compare our method with the previous best-performing model, MTMamba, using two different encoders: Swin-Base Transformer (denoted as '-Base') and Swin-Large Transformer (denoted as '-Large'). The results suggest that models with greater capacities typically exhibit superior performance. Furthermore, our approach has demonstrated superior performance across all encoder variants.

**Qualitative Visualization.** We qualitatively compared our proposed MSM with the previous best-performing method, as shown in Figure 6. Our method shows clear improvements in detail, as highlighted in the circled regions. For more visual comparisons, please refer to Appendix A.3.

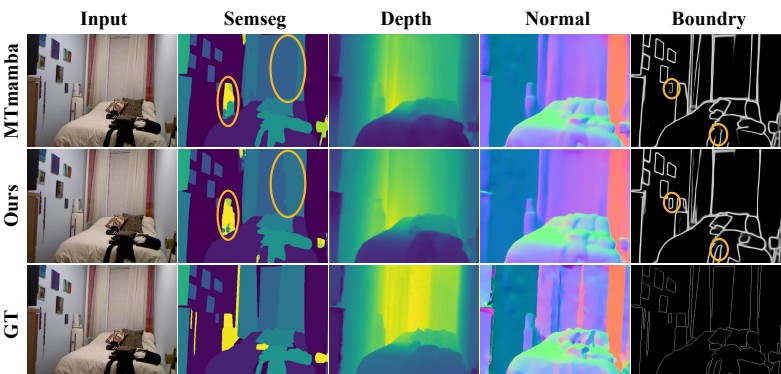

Figure 6: Qualitative comparison with the best performing method on NYUD-v2. Our method generates better multi-task prediction details.

## 5 CONCLUSION

We proposed a Multi-Scale Mamba (MSM) framework for addressing the diverse preferences of scene structure granularity for different tasks in multi-task dense prediction. We introduce a multi-scale scanning mechanism (MS-Scan) that comprehensively constructs scene structure information at various scales. Additionally, we build two multi-scale Mamba modules (TS-MSM and CT-MSM) that meet the diverse needs of task representation construction, thereby alleviating the difficulty of feature learning in multi-task dense scene prediction. Both qualitative and quantitative results show that our method significantly enhances performance.

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

# A APPENDIX

## A.1 FEATURE MERGE DETAILS OF CT-MSM

In CT-MSM, we have adapted the feature fusion approach from MTMamba (Lin et al., 2024) to merge multi-scale task-share features $\mathcal{F}_{ms}^s$ and task-specific features $\mathcal{F}_t^s$, as illustrated in the figure 7. The core is to generate selection values $\{\mathcal{G}_t^s\}_{t=1}^T$ for weighting the task-shared $\mathcal{F}_{ms}^s$ and task-specific features $\{\mathcal{F}_t^s\}_{t=1}^T$ to obtain the fused task features $\{F_t^s\}_{t=1}^T$ for all tasks. In implementation, before weighting the task-specific features, they undergo further refinement through convolution and SS2D (Liu et al., 2024) operations.

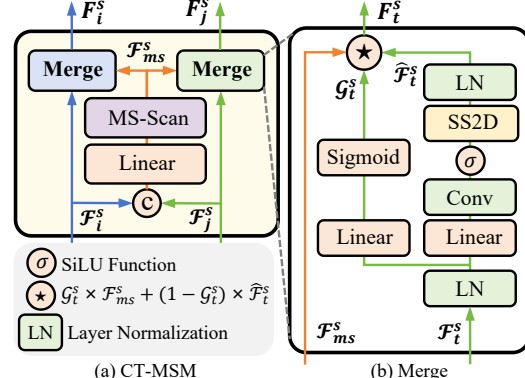

Figure 7: Merge Details of CT-MSM.

$$\mathcal{F}_{t,LN}^s = \mathrm{LN}(\mathcal{F}_t^s), \tag{11}$$

$$\mathcal{G}_t^s = \mathrm{Sigmoid} \circ \mathcal{P}(\mathcal{F}_{t,LN}^s), \tag{12}$$

$$\hat{\mathcal{F}}_t^s = \mathrm{LN} \circ \mathrm{SS2D} \circ \mathrm{SiLU} \circ \mathrm{Conv} \circ \mathcal{P}(\mathcal{F}_{t,LN}^s), \tag{13}$$

$$\mathrm{F}_t^s = \mathcal{G}_t^s \times \mathcal{F}_{ms}^s + (1 - \mathcal{G}_t^s) \times \hat{\mathcal{F}}_t^s, \tag{14}$$

where LN denotes the Layer Normalization, $\mathcal{P}$ represents a linear projection, Sigmoid and SiLU are the sigmoid function and SiLu function respectively, Conv($\cdot$) is the convolution layer.

## A.2 LIGHTWEIGHT MSM

MSM framework introduces minimal additional computational cost (**+0.01 GFLOPs**) compared to the original MTMamba, yet achieves significant performance improvements. To further validate the effectiveness of our method, we present Dilated Multi-Scale Mamba (DMSM), a lightweight version of MSM, which achieves superior performance with reduced computational complexity compared to MTMamba. DMSM conducts sparse scanning within each scan branch $\mathcal{B}$. Specifically, as shown in Figure 8, we perform dilated sampling in generating multi-scale sequences from image features instead of using all tokens. When restoring sequences to image features, we perform linear interpolation. These operations do not introduce any parameters and exhibit a reduced computational burden due to sampling a subset of tokens for modeling. Experimental results, as depicted in Table 6, demonstrate the effectiveness of meeting the diverse requirements of tasks for scene granularity in multi-task dense prediction. Among them, **FLOPs$^m$** denotes the complexity of SSM operations, **FLOPs$^o$** is the complexity of other operations, and **FLOPs = FLOPs$^m$ + FLOPs$^o$** the total complexity. All experiments utilize Swin-Large Transformer as encoder.

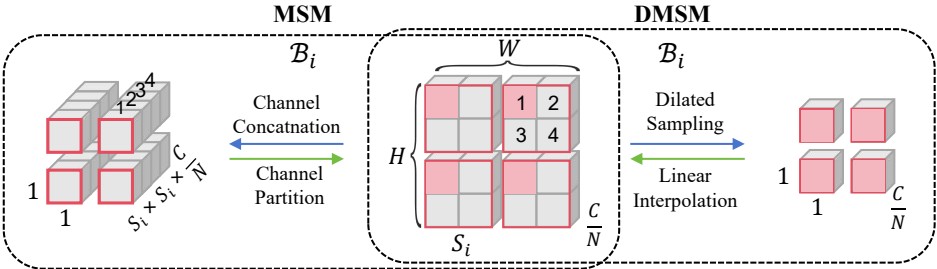

Figure 8: Comparision of MSM and DMSM.

Table 6: Performance Comparison of Dilated Multi-Scale Mamba (DMSM) and MTMamba.

| Model | Semseg mIoU ↑ | Depth RMSE↓ | Normal mErr ↓ | Boundary odsF ↑ | MTL Gain $\Delta_m$ ↑ | FLOPs$^m$ (G)↓ | FLOPs$^o$ (G)↓ | FLOPs (G)↓ | # Params (M)↓ |
|---|---|---|---|---|---|---|---|---|---|
| STL Model | 54.32 | 0.5166 | 19.21 | 77.30 | +0.00 | - | - | 1074.79 | 888.77 |
| MTMamba | 55.82 | 0.5066 | 18.63 | 78.70 | +2.38 | 81.72 | 459.09 | 540.81 | 307.99 |
| DMSM (Ours) | 56.95 | **0.4813** | 18.64 | 78.90 | +4.18 | **60.50** | **450.57** | **511.07** | **307.99** |
| MSM (Ours) | **57.79** | 0.4832 | **18.63** | **79.00** | **+4.51** | 81.72 | 459.10 | 540.82 | 396.54 |

## A.3 MORE VISUAL COMPARISON RESULTS

**Task Attention.** To compare task attention against the state-of-the-art method, we visualize the results in Figure 11. Our method demonstrates a more precise attention range across all tasks, aligning with the intrinsic requirements of specific tasks. Specifically, it accurately captures task-specific object relationships, thereby improving overall performance and scene understanding. These findings suggest that the incorporation of multi-scale scanning in MSM addresses the varying demands for scene structural granularity across distinct tasks, thereby mitigating the difficulties in feature learning within multi-task dense prediction and improving the alignment of task-specific features with the intrinsic requirements of each task.

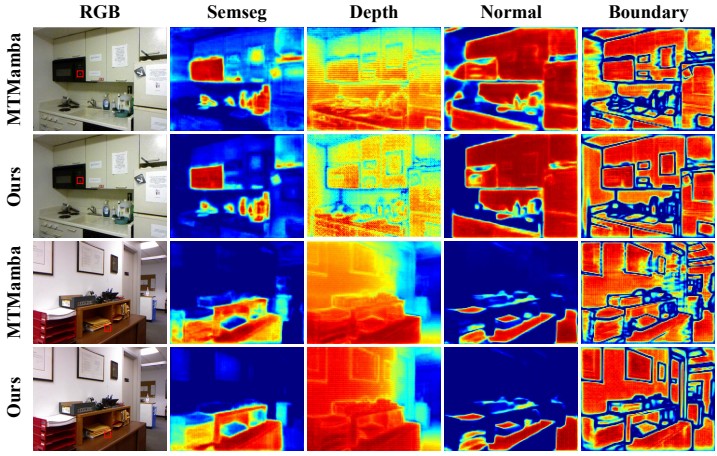

Figure 9: More task attention comparison on NYUD-v2 dataset.

**Qualitative Comparison.** We present more qualitative results compared with the SOTA methods, MTMamba (Lin et al., 2024). In Figure 11 - Figure 13, we can see that our method generates better multi-task prediction details, highlighted in the circled regions.

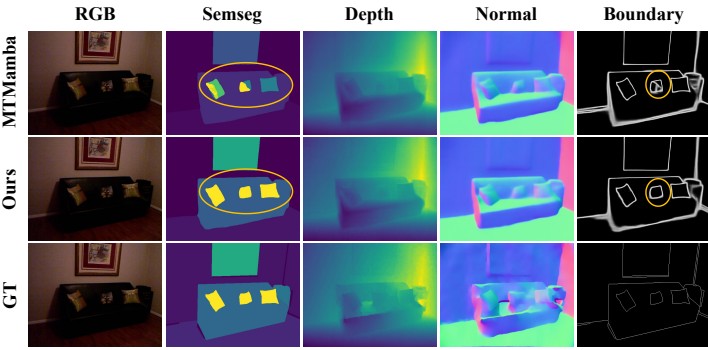

Figure 10: More Qualitative comparison on NYUD-v2 dataset.

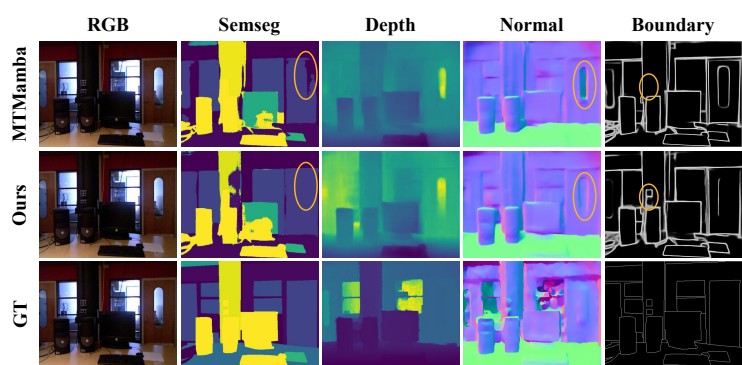

Figure 11: More Qualitative comparison on NYUD-v2 dataset.

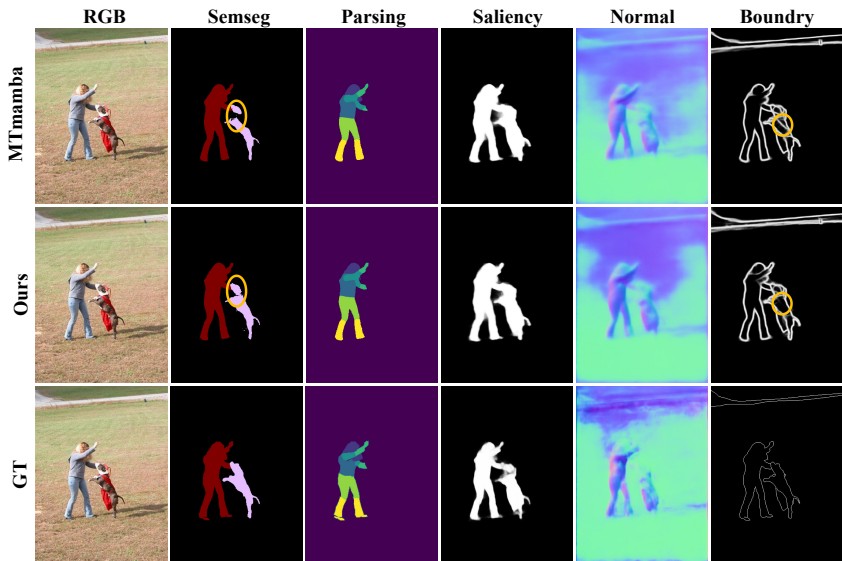

Figure 12: More Qualitative comparison on Pascal-Context dataset.

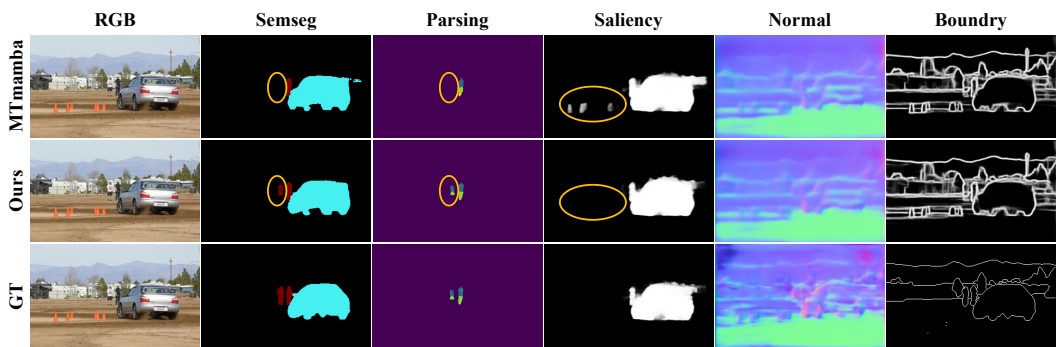

Figure 13: More Qualitative comparison on Pascal-Context dataset.

