# OpenReview forum: "MSM: Multi-Scale Mamba in Multi-Task Dense Prediction"
_ICLR.cc/2025/Conference — ICLR 2025 Conference Withdrawn Submission_

### Official Review · Reviewer_yzK9 · 2024-10-28

**Soundness:** 3
**Presentation:** 3
**Contribution:** 3
**Rating:** 5
**Confidence:** 5

**Summary:**

This paper proposes an approach, Multi-Scale Mamba (MSM), for multi-task dense prediction. Specifically, the authors employ Multi-Scale Scanning (MS-Scan) to establish global feature relationships at various scales for a comprehensive visual representation.The authors adaptively merge task-shared information from multiple scales across different task branches. The proposed method is evaluated on two datasets and the experiment results also achieve comparable performance.

**Strengths:**

S1 The proposed architecture is possibly novel in terms of how low-rank experts are used for multiple tasks.

S2 It outperforms the state-of-the-art in most cases.

S3 The organizational structure of the paper is clear and reasonable.

**Weaknesses:**

W1. The primary concern with the paper is the lack of novelty. Multi-scale approaches have been extensively explored in MTL, and the application of such methods to MTL, while valuable, does not contribute significantly to new knowledge or innovative techniques in the field. The "Multi-Scale Mamba" proposes a combination that, though functional, does not distinctly differentiate itself from existing methods (e.g., MTmamba) in terms of conceptual innovation.

W2. This paper states: this architecture refines the task-generic features G obtained from the encoder into task-specific features F. How to get multiple F features from one task-generic G feature at the same scale? When G is replicated T times for each task, why can the replicated features be called task-specific features?

W3. When the number of tasks increases (more than 2 tasks), the features after channel-wise concatenation in CT-MSM are very large (e.g., three tasks in MFR block1, 4C+4C in TS-MSM; 4C+4C+4C in CT-MSM). That's a lot of computation. Is this an appropriate approach that can be computationally expensive as the number of tasks grows?

W4. Some of the confusion comes from the size of the features, i.e., F_{in}^s, F_{h}^s and F^s in the MFR block.

W5. CNN-based MTL and Transformer-based MTL methods use HRNet and ViT or Swin-Transformer as backbones. The proposed MSM model is classified in mamba-based methods, but MSM use Swin-Transformer as backbone. Did the authors consider using Mamba-based (i.e., Vision mamba, EfficientVMamba, VMamba) as backbone?

W6. More details about the experimental results are forgotten in Tables 1 and 2, including the parameters, flops and $\Delta_m$ of the model. The authors are encouraged to add more recent MTL methods (e.g., TaskExpert, DiffusionMTL) as a comparison.

W7 We look forward to the authors releasing the code and models that will benefit the MTL community.


W8 Minors:
 … CT-MSM module. these modules enhance the model’s … -> … CT-MSM module. These modules enhance the model’s…
In Eq (1), the mathematical notation “\circ” needs to be explained in detail.

**Questions:**

Q1 In Table 3, why would the values of the STL model be lower than those of Baseline model (i.e., MT model)? Such a higher value of MT than ST is very inconsistent with the common sense of multi-task learning.

---

### Official Review · Reviewer_maU8 · 2024-11-03

**Soundness:** 3
**Presentation:** 3
**Contribution:** 3
**Rating:** 5
**Confidence:** 4

**Summary:**

The paper presents a framework called Multi-Scale Mamba (MSM) for multi-task dense prediction in multi-task dense prediction. The MSM framework aims to address the challenge of diverse granularity requirements for scene structure across different tasks by introducing a multi-scale scanning mechanism (MS-Scan) and two specialized multi-scale Mamba modules: Task-Specific Multi-Scale Mamba (TS-MSM) and Cross-Task Multi-Scale Mamba (CT-MSM). These components enhance the model's ability to deliver comprehensive visual representations and meet the varied demands of tasks, leading to improved performance in multi-task dense prediction scenarios.

**Strengths:**

1. The MSM framework adapts to the diverse granularity demands of various tasks, which is crucial for multi-task dense prediction where each task may require different levels of scene structure detail.

2. Experiments on NYUD-V2 and PASCAL-Context datasets demonstrate MSM's superiority over state-of-the-art methods.

3. The MSM framework introduces minimal additional computational cost compared to the original MTMamba.

**Weaknesses:**

1. The motivation is not very strong. I have not seen the authors discuss thoroughly why we need the mamba for multi-task dense prediction tasks. It is like combining the mamba with the multi-task dense prediction, rather than the dense prediction needs mamba.

2. As mamba is superior to Transformer in efficiency, I have not seen any comparison between the MSM and Transformer-based models in computational cost, efficiency and speed analysis.

**Questions:**

1. Line 101 These->these.

2. I would suggest the authors to discuss limitations of the proposed method.

---

### Official Review · Reviewer_9Wxg · 2024-11-08

**Soundness:** 3
**Presentation:** 3
**Contribution:** 2
**Rating:** 5
**Confidence:** 4

**Summary:**

This paper proposes Multi-Scale Mamba (MSM), which extends MTMamba with a task-aware hierarchical scene modeling function. It utilizes a MS-Scan mechanism and two specialized multi-scale Mamba modules to build thorough visual representations, and accommodate the different granularity needs of each task.
Experimental results show that MSM outperforms leading methods on multi-task dense prediction.

**Strengths:**

1. The model structure and illustrations are thorough and well-explained.

2. This manuscript shows clear and detailed visualizations effectively illustrate the model's performance compared with previous work.

**Weaknesses:**

1. The novelty is not enough. Using multi-scale features is a common approach in dense prediction tasks. This paper has no true innovation in the scanning strategy itself.

2. Does the introduction of multi-scale scanning lead to extra computational cost? Will it impact the model’s inference speed?

3. All qualitative analyses are based on positive cases. Could the authors provide failure case visualizations and analyze if multi-scale processing affects local representation?

4. The results in Tables 1 and 2 seem quite minimal, raising doubts about the limited effectiveness of the proposed approach.

5. Typos: the bolded value in the 2nd column of Table 3 is not the best result.

**Questions:**

None

---

### Official Review · Reviewer_Nf7z · 2024-11-08

**Soundness:** 2
**Presentation:** 3
**Contribution:** 2
**Rating:** 3
**Confidence:** 4

**Summary:**

In this paper, the authors propose using Mamba blocks for the multi-task dense prediction task. There are mainly two modules proposed, which are Task-Specific Multi-Scale Mamba (TS-SMS) and Cross-Task Multi-Scale Mamba (CT-MSM). The authors use the experiments to prove the effectiveness of the proposed model.

**Strengths:**

1. In the comparison (Table 1 and Table 2), the performance of the proposed method is better than previous methods. The results indicate that the proposed method consistently outperforms existing techniques across multiple tasks.
2. The authors do some ablation studies on the module design, which further shows the effectiveness.
3. The authors provide some visualization results. These visual results make it more accessible and easier to interpret for readers.

**Weaknesses:**

1. The comparison methods are not well selected. I have investigated several comparison methods, such as InvPT, MQTransformer, and ATRC (except for MTMamba). These methods are relatively outdated and have lower computational costs compared to current state-of-the-art techniques. By choosing older methods with lower computational demands, the evaluation may not accurately reflect the performance of more advanced algorithms. I hope the authors can provide a detailed comparison of the computational cost of this method with previous ones since the authors think one of the biggest advantages of using Mamba is computational complexity (L049).

2. Compared to MTMamba, the contribution is quite limited. The proposed two modules are like to use Mamba blocks to fuse the multi-scale features. This is a common idea of dense prediction. The advantages and disadvantages are very clear: the advantage is that it can improve performance, but the disadvantage is that it will greatly increase the computational costs. In order to better study the contribution of this article, I think the authors should at least compare it with multi-scale and transformer block and CNN block. Note that the SS2D op and MTMamba are not the contributions of this paper.

In conclusion, I think this paper lacks a comparison with recent state-of-the-art methods and a detailed analysis of efficiency. Also, the contribution is not clear. Considering that MTMamba already uses Mamba for this task, the new stuff in this paper is only the multi-scale based on my understanding.

(Minor)
L153: Pineline -> Pipeline
L049: the -> The

**Questions:**

What is the difference between TS-MSM and ST-MSM? I think it might be a typo, but there are so many of both of them.

---

### Note · Authors · 2024-11-26

I have read and agree with the venue's withdrawal policy on behalf of myself and my co-authors.